# Blood Flow Restriction Is Not Useful as Soccer Competition Recovery in Youth Male National-Level Soccer Players: A Crossover Randomised Controlled Trial

**DOI:** 10.3390/sports11050099

**Published:** 2023-05-07

**Authors:** Christian Castilla-López, Natalia Romero-Franco

**Affiliations:** 1RCD Mallorca, E-07011 Palma de Mallorca, Spain; christiancastillalopez@gmail.com; 2Nursing and Physiotherapy Department, University of the Balearic Islands, E-07122 Palma de Mallorca, Spain; 3Health Research Institute of the Balearic Islands (IdISBa), E-07120 Palma de Mallorca, Spain

**Keywords:** blood flow restriction therapy, athletic performance, soccer, fatigue, recovery, wellness

## Abstract

In soccer, blood flow restriction (BFR) is used to optimise between-match recovery. However, the benefits are unclear. This study evaluated the effects of BFR as a recovery strategy after a competition on countermovement jump (CMJ) height, rating of perceived exertion (RPE) and the wellness of soccer players. Forty national-level soccer players were allocated into two conditions: BFR (an active recovery session wearing a BFR device, 24 h after a competition) or NoBFR (the same recovery without BFR). CMJ, RPE and wellness were evaluated the day (CMJ and RPE) or the morning (wellness) before the competition; just after the competition (CMJ and RPE); and 24, 48 (wellness) and 72 h later. After 4 weeks, the players changed conditions. All players showed impaired CMJ (*p* = 0.013), RPE (*p* < 0.001) and wellness (*p* < 0.001) after the match compared with the baseline. The CMJ returned to the baseline 24 h later and wellness returned 48 h later. Only in the BFR condition did the RPE remain impaired 24 h after the match, which was also the moment after finishing the BFR recovery session (*p* < 0.001). BFR during active recovery does not provide any additional benefits compared with traditional exercise modalities to recover CMJ, RPE and wellness in youth national-level soccer players. BFR could even induce an immediate higher RPE.

## 1. Introduction

In elite soccer, the players must play up to three competitive matches per week in European leagues due to social demands and soccer monetisation [1]. Because the recovery time is becoming shorter and shorter, optimising recovery is crucial to maintain optimal performance [1,2] and to prevent additional injuries [3,4]. Consequently, sports professionals implement active post-match strategies to accelerate the recovery process in terms of psychological and physical parameters [4]. Jogging, submaximal sprints and small-sided games (SSGs) are frequently considered for active recovery in a tactical soccer periodization schema [5,6].

Recently, adding blood flow restriction (BFR) to the aforementioned recovery routines has become popular among professional soccer teams and for other team sports [7]. It consists of applying adjustable cuffs to the limb to be trained and through a sphygmomanometer, controlling the blood pressure [8]. This technique has traditionally been combined with resistance training at a low or moderate intensity [9] to accelerate the injury rehabilitation procedure. This application is often justified due to the effects of this technique to optimise the supply of oxygen (O_2_) and nutrients, and to remove metabolic by-products from active muscles [10]. This same explanation has motivated sports professional to include it as part of the recovery strategy between competitive matches [7].

It seems that, combined with aerobic exercises, BFR could be an alternative to traditional passive therapies such as cryotherapy, massage or pressotherapy [11]. The potential regenerative character of this technique is based on its effects on vasodilatation and the O_2_ supply to induce faster restoration of the stretch-shortening cycle (SSC) [12]. Ischemic preconditioning (IPC) or post-exercise ischemic conditioning (PEIC) consisting of short rest periods of intermittent vascular occlusion of the lower limbs has been proved to benefit performance recovery. The schema frequently used by other authors is based on two to five sets × 2–5 min separated by 5 min of reperfusion with a pressure of 50 or 220 mmHg. [7]. Daab et al. [13] suggested that applying PEIC using a schema of 3 × 5 min separated by 5 min of reperfusion at 50 mmHg of the systolic blood pressure (SAP) after a simulated soccer match could contribute to the faster recovery of neuromuscular functions by decreasing muscle soreness and muscle damage in semi-professional soccer players. Similarly, Patterson et al. [14] showed the benefits of IPC prior to exercise-induced muscle damage (EIMD) regarding the recovery time of the maximum involuntary contraction (MIVC) in healthy males. On the contrary, Gibson et al. [15] indicated that the IPC protocol did not benefit the sprint performance of team sports players. The possible mechanisms associated with IPC or PEIC may be increased blood flow [16], reduced oxidant production [17], higher adenosine levels [18] and decreased inflammatory responses [19]. However, there is not a standardised IPC or PEIC methodology and a simulated match may not require the same physical and psychological demands as a real competitive match. To our knowledge, there are no studies that have applied BFR during active recovery as a post-competition strategy in soccer players. Therefore, studies need to confirm these potential regenerative effects to determine whether the practical applications after a competitive soccer match can avoid muscle soreness and fatigue. The purpose of this study is to evaluate the effects of BFR as strategy to recover jumping ability, subjective fatigue and wellness perception in youth male national-level soccer players after a competitive soccer match. We hypothesise that active recovery including BFR decreases the time to recover for the aforementioned variables compared with the same recovery strategy without BFR.

## 2. Materials and Methods

### 2.1. Design

A crossover randomised controlled trial with two conditions was designed: BFR and NoBFR. For the BFR condition, the participants wore the BFR device during an active recovery session 24 h after a competitive soccer match. For the NoBFR condition, the participants carried out the same recovery session but without BFR. To ensure a similar physical and psychological status among the players, only those who played 50 min or more in the previous competitive match were included. The following variables were assessed in all soccer players before and after the competitive match: the maximal vertical jump ability based on countermovement jump (CMJ), subjective fatigue by rating perceived exhaustion and wellness perception assessed with the Wellness Questionnaire (WQ). Four weeks later, the players repeated the entire process with the other condition (Figure 1). The players were randomly assigned to the BFR or NoBFR condition to establish the intervention order. For randomisation, the web-based randomizer.org program was used. The competitive soccer matches were previously selected to ensure four weeks between both conditions, but also considering a similar level in the opponent soccer team. In all cases, the competitive soccer match ended in a tie. Testing, the competitive soccer match and the recovery sessions were completed in the sport facilities of the soccer teams. This study took place during the 2021–2022 soccer season of Youth League and Honour Spanish Division teams.

### 2.2. Participants

The sample size was calculated with G*power version 3.1.9. Considering an alpha level of 0.05, a beta risk of <0.2 in a bilateral contrast and 10% as the possible subject loss, a minimum of 30 participants were required to identify a statistically significant difference (*p* < 0.05, power level of 0.8) for vertical jump as the primary outcome variable. The inclusion criteria included: participants who belong to a soccer team from the Spanish Youth Soccer League (Youth League and Honour Spanish Division), at least 5 years of soccer experience and no previous experience with wearing a BFR device during exercise. Furthermore, the players had to be included in the starting line-up for the match during the intervention period. The exclusion criterion was the presence of injuries within 3 months. In addition, goalkeepers were excluded because their fatigue was not comparable to the rest of the player positions (defenders, midfielders and forwards). All regional soccer teams that participated in the Spanish Youth Soccer League (*n* = 6) were contacted by mail by the research staff. Only three soccer teams agreed to participate in the study. Thus, 40 youth male national-level soccer players (age: 17.06 ± 0.77 years; body mass: 70.16 ± 6.96 kg; height: 1.77 ± 0.06 m; body mass index: 22.20 ± 1.93 kg/m^2^; minutes played during match: 79.85 ± 14.06 min) voluntarily participated in the study. During the study period, the players regularly performed five weekly soccer training sessions with their team. The weekly programme was similar for all players and consisted of a tactical periodization model for soccer, with one official match per week [6]. The study was conducted according to the Declaration of Helsinki. All participants received detailed information about the study and signed the informed consent form. For underage athletes, a parent or legal guardian also signed the informed consent form.

### 2.3. Testing

Before the start of the study, all players completed a questionnaire to gather information about sociodemographic and medical characteristics (age, years of experience in soccer and previous injuries). Height and body mass data were collected for all players using a portable height rod Seca^®^ 213 (Hamburg, Germany) and a Tanita BC-731 scale (Dongguan, China), respectively. All evaluations were completed at a similar time (3:30–4:30 p.m.), with similar climatological conditions. Prior to testing, players received instructions to abstain from strenuous activity outside of the proposed training or competition during the intervention period (from 24 h before testing to 72 h after the match) and were encouraged to maintain their normal dietary and fluid intake habits during the study. The testing sessions were always carried out under the supervision of the fitness coach, who was blinded to the participants’ group assignments.

#### 2.3.1. Countermovement Jump (CMJ) Test

The CMJ test was developed to evaluate neuromuscular function. All soccer players started in an upright position with their hands contacting their hips before executing a quick flexion–extension movement of the legs and a maximum vertical jump. Three non-consecutive jumps were performed, with 45 s of rest between the trials [20]. The participants were instructed with the command ‘3, 2, 1, Jump’. Moreover, verbal encouragement was provided to ensure maximal effort in each jump. A force platform (ForceDecks FDLite-7735, Vald Performance, Brisbane, Australia) was used to obtain the average jump height (cm), in line with a previous study [21]. We selected jump height, one of the most important metrics supported by the literature, due to the complex study protocol and the close measurement times and to avoid extending the evaluation time [22]. Measurements were taken as follows: 24 h before the competitive soccer match (Pre24h), just after the competitive soccer match (PostMatch), 24 h after the competitive soccer match (and just after the active recovery session) (Post24h) and 72 h after the competitive soccer match (Post72h). Because the weekly programme established a rest day 48 h after the competitive soccer match, CMJ was not evaluated at this time. To ensure the correct execution of the test, all soccer players performed a standardised warm-up consisting of 5 min of jogging on a treadmill (Salter Terra PT1750, Barcelona, Spain) at 9 km/hour, 5 min of active mobility, 5 min of running exercises similar to FIFA11+ [23] and progressive CMJ. On match day (MD), a warm-up was not needed because CMJ was assessed just after the competitive soccer match [24]. We used a sampling rate of 1000 Hz to detect the beginning and the end of the movement at a 30 N deviation from the starting body weight. We employed ForceDecks software (Vald Performance, Brisbane, Australia) to analyse the data and the force–time curve to determine the CMJ height derived from the flight time [25].

#### 2.3.2. Rating of Perceived Exertion (RPE)

RPE was evaluated using the modified Borg Scale (from 0 points [no fatigue at all] to 10 points [maximal fatigue]) [26]. It was completed 30 min after the end of the training sessions or competitive soccer match [27] as follows: Pre24h, PostMatch, Post24h and Post72h [28]. Because the players had a rest day 48 h after the competitive soccer match (Post48h), RPE was not evaluated at that time. A comparable training load was ensured by evaluating the RPE and multiplied by the duration of the training session according to other studies [29]. Additionally, the external load was measured using GPS data (distance covered in metres) collected with a device that the players wore during the match and training sessions.

#### 2.3.3. Wellness Questionnaire (WQ)

The WQ comprised five items: fatigue, sleep quality, general muscle soreness, stress level and mood. Each item was evaluated according to a 5-point scale (from 1 point [worst status] to 5 points [best status]) [30]. The sum of the scores from the five items was the Hooper index, which was used to measure stimulus-induced fatigue in each player (with 25 points as the best well-being status) [31]. The participants always completed the questionnaire in the morning, 2 h before beginning the training session or competitive soccer match, as follows: on MD, Post24h, Post48h and Post72h, similarly to previous studies [32,33].

### 2.4. Interventions

Two weeks before the start of the study, all players completed a familiarisation process consisting of one session wearing the BFR device during active recovery using the specific interventions explained below.

#### 2.4.1. BFR Active Recovery Session (BFR Condition)

The participants completed a recovery session while wearing the BFR device 24 h after the competitive soccer match. The participants wore an Occlusion Cuff^®^ (Belfast, United Kingdom) with a wrap size (7 × 82 cm in length) that restricted the blood flow with a mechanical manometer. The cuffs were placed below the gluteal line in both legs and inflated at the same time. The upright position was used to set the blood pressure to ~60% of the limb occlusion pressure (LOP) [34]. The arterial blood flow occlusion was determined using the following formula for the lower limb:
$$\begin{array}{c} Lower\;limb\;arterial\;occlusion\;(mmHg)=5.893\times (thigh\;circumference)+0.734\times \\ (diastolic\;blood\;pressure)+0.912\times (systolic\;blood\;pressure)-220.046 \end{array}$$


Between drills, the elastic wraps were deflated for 90 s [35]. The recovery session consisted of a warm-up with running (5 min), six sprints from one box to the other box of the field at 60–70% of a maximum repetition measured during the preseason (jogging from the box to the middle and walking to the other box) and an interval drill in the form of rounds with the ball, nine versus two players (three sets of 5 min, with 90 s of rest between each set). Appendix A shows an example of these exercises. Prior to the recovery session, an activation part was executed following the recommendations of FIFA11+ [23].

#### 2.4.2. No-BFR Active Recovery Session (NoBFR Condition)

The participants completed the same recovery session as the BFR condition, but they did not wear the BFR device.

### 2.5. Statistical Analyses

Descriptive data were presented as the mean ± standard deviation (SD). The normality of the data was checked using the Shapiro–Wilk test. Repeated measured analysis of variance (ANOVA) was used to explore within- and between-group differences for CMJ, RPE and the WQ items. The assumption of sphericity was checked using Mauchly’s test. When the assumption of sphericity was not met, Greenhaus–Geisser adjusted values and the degrees of freedom test were used. In the case of a significant F-test, the Bonferroni test for multiple comparisons was applied. To determine the magnitude of between-group differences, the 95% confidence interval (CI) and the calculated effect size (ES) were determined using Cohen’s d, interpreted as small (d ≤ 0.2), moderate (0.2 < d ≤ 0.8) or large (d > 0.8). SPSS Statistics version 21.0 (IBM Corp., Armonk, NY, USA) was used for all statistical analyses. Statistical significance was set at *p* < 0.05.

## 3. Results

The RPE, CMJ and WQ results are shown in Table 1 for the BFR and NoBFR conditions. There were no significant between-group differences before and after the competitive soccer match, except for RPE at Post24h. The repeated measures ANOVA did not show a significant group effect or a group-by-time interaction, except for RPE, which showed a significant group-by-time interaction (F = 3.695, *p* = 0.018). There were significant time effects for all of the variables.

Concerning the RPE, the perceived exhaustion increased in all players at PostMatch (*p* < 0.001). These values decreased at Post24h (*p* < 0.001), but were significantly higher for the BFR group compared with the NoBFR group (*p* = 0.002). The RPE increased at Post72h for both conditions (NoBFR *p* < 0.001; BFR *p* = 0.005), but this increase was greater for the NoBFR group (*p* = 0.026) (Figure 2).

Regarding the CMJ, all players, regardless of the group, showed a decreased jump ability at PostMatch (*p* = 0.014). Although the values remained low at Post24h, there were no significant differences compared with the baseline (*p* > 0.05). At Post72h, all players increased their jump ability compared with PostMatch (NoBFR *p* = 0.001, BFR *p* = 0.003) and compared with Post24h (NoBFR *p* < 0.001, BFR *p* = 0.015) (Figure 2).

Regarding the WQ, the Hooper index significantly decreased at Post24h for both conditions (*p* < 0.001). Although this score improved at Post48h, the values remained lower than those observed prior to the match (*p* = 0.010). At Post72h, the values were similar to the baseline (*p* > 0.05). Regarding the WQ sub-items, there were no significant between-group differences (*p* > 0.05), but there were time effects for fatigue, sleep and muscle soreness. All players, regardless of the conditions, showed decreased values in these three sub-items at Post24h compared with the baseline (fatigue *p* < 0.001; sleep *p* < 0.001; muscle soreness *p* < 0.001). In the case of fatigue, the values were also lower at Post48h compared with those observed prior to the match (*p* = 0.015), but were similar to the baseline at Post72h (*p* > 0.05) (Figure 2). The sub-item data are presented in Appendix A.

## 4. Discussion

The main finding of this study was that wearing a BFR device during an active recovery session 24 h after a competitive soccer match did not provide beneficial effects to recover jumping ability or perceived wellness. Only players who wore the BFR device had increased perceived exhaustion immediately after the recovery session. For the other variables, the fluctuations were similar for both conditions from 24 h before the competitive soccer match up to 72 h after the match.

To the best of the knowledge of the authors, researchers have not evaluated the methodology used in this research. Hence, it is difficult to compare our data to the literature. The previous studies that investigated BFR in endurance athletes either cyclically (i.e., during repeated sprint ability exercises, during lower intensity intervals, etc.) or continuously (i.e., during strength training) reported physiological responses of fatigue and performance in different populations [12]. Although Held et al., who evaluated the effects of rowing at a low intensity with BFR for 5 weeks, observed increased VO2max (maximal oxygen consumption) in elite rowers [36], most of the studies have not reported benefits in terms of performance or fatigue recovery when BFR is used. These studies suggest that the mechanical stress, contractile capacity or protection of cerebral function as key factors that could condition the beneficial effects [7,37]. Mechanical stress has been related to a higher heart rate and ventilation during aerobic exercise with BFR [38]. Contractile capacity is affected by BFR due to venous distension and metabolic accumulation, increasing the nociceptive group IV muscle afferents [39]. The protection of cerebral function under BFR could be linked to the spinal motoneurons reducing muscle activation and provoking central fatigue [40].

As other authors have suggested, the pain induced while wearing a BFR device could be detrimental to the motivation of the players to continue exercising [41]. Our results were in line with this explanation: we observed a temporal detrimental effect from wearing the BFR device on perceived exhaustion. In agreement, Williams et al. [42] applied intermittent lower limb occlusion with PEIC (2 × 3 min at 176–266 mmHg) and reported no positive effects compared with the sham group (2 × 3 min at 50 mmHg) on CMJ height after sprinting in academy rugby players at 24 h. The authors suggested that the training status could be an additional reason mediating the efficacy of vascular occlusion on physical performance [42]. Although an immediate increase in RPE when players wore the BFR device was observed, this negative affect did not occur for the other variables at the subsequent time points. It is important to note that because the rest day was established 48 h after the competitive soccer match, we could not assess RPE or CMJ 24 h after the recovery session.

Another result to highlight was the perceived exhaustion at Post72h, regardless of the recovery session that the players completed. At this time, the perceived exhaustion increased to ~6 points, which was a midpoint between ‘heavy’ and ‘very heavy’. This result was expected because the RPE was always evaluated 30 min after the training session. Moreover, the high RPE values were common on that day due to the progressive increment and distribution of the load during the week. The results of the present study agreed with those reported by other researchers who evaluated professional soccer players with a similar periodization training method [43]. The potential influence of the timing in the season when our study was completed should be highlighted, due to the congestion of the competitive soccer matches.

In the present study, the CMJ height was measured to know the maximal jumping ability due to its sensitivity, reliability and validity in team sports, as corroborated in a previous study [21]. The CMJ height was immediately affected due to the competitive soccer match in all players. After this time, the CMJ height increased to values comparable to 24 h before the match, a finding that was similar to a previous study [44]. CMJ height is a sensible value to monitor neuromuscular fatigue after a soccer match [44]. In the present study, the CMJ height results confirmed that the players reached similar neuromuscular fatigue and performance following the soccer competition. Thus, all soccer players had a comparable neuromuscular fatigue status before the recovery session, regardless of the condition they were submitted to each time. Once the recovery session was completed, with or without BFR, all soccer players showed similar recovery in the vertical jump. The CMJ performance presented in the BFR group was consistent with the study by Daab et al. [13] at 72 h after PEIC (3 × 5 min separated by 5 min of reperfusion at 50 mmHg) in semi-professional soccer players compared with a placebo group. The values at Post72h were comparable to those reached 24 h before the soccer competition. Page et al. [45] indicated that PEIC (3 × 5 min separated by 5 min of reperfusion at 220 mmHg) effectively accelerated vertical jump height after EIMD in healthy, recreationally active men. In contrast, Northey et al. [46] showed a decrease in the jumping ability of strength-trained men using PEIC (2 × 3 min separated by 3 min of reperfusion alternating in each leg at 220 mmHg) 24 h after fatiguing resistance exercise. There have been controversial effects regarding jump height and neuromuscular function presented by several authors, a phenomenon that has contributed to the current inability to clarify the hypothetical associated mechanisms. These possible mechanisms might attenuate oxidative stress by reducing mitochondrial reactive oxygen species (ROS) production. This reduction in ROS could prevent oxidative stress in muscle cells [46]. An increase in nitric oxide could simultaneously decrease ROS, enhancing performance and reducing muscle damage and neuromuscular fatigue [46]. Moreover, attenuation of the inflammatory response is linked to downregulation of circulating leucocytes [19].

Regarding the WQ, the Hooper index was immediately impaired after the soccer competition and remained affected 24 h later. The psychological effect reflected in the WQ in this study was consistent with the neuromuscular fatigue reflected in the CMJ height. Nedelec et al. [47] reported a similar finding during a standard soccer training week. Indeed, according to Silva et al. [48], jump performance was not recovered despite the fact that sprint ability was recovered at 72 h after a match. These authors also affirmed that recovery might depend on the individual characteristics of the players, suggesting that the psychological parameters need more than 72 h to be completely recovered.

When we specifically analysed the five sub-items in the WQ, there were no differences between the conditions. In all players, muscle soreness, fatigue and sleep quality were negatively affected the morning after the competitive soccer match, while mood and stress were unaffected. These results were consistent with those observed by Deely et al. [33] after a strenuous training session. These authors found that fatigue, muscle soreness and readiness to train were the most affected perceptual items in academic-level soccer players. The muscle soreness results did not conform to the theory of reducing muscle pain by increasing the NOS (nitric oxide synthase) level [49]. Daab et al. [13] reported opposite results from the present study, and in agreement with the aforementioned theory. They found lower muscle soreness in soccer players at 24 h after a simulated soccer match. Subjective fatigue was the sub-item that took the longest to be completely recovered, perhaps because of the metabolic accumulation of BFR [39]. Moreover, the strenuous training session could have induced greater neuromuscular and physical fatigue than the competitive soccer match considered in the present study [33].

Despite the fact that the competitive soccer match induced similar neuromuscular fatigue and subjective perception of fatigue for all players, the results did not support the initial hypothesis for any of the measured variables. It seems that the physiological and metabolic benefits that make BFR an appropriate method to accelerate injury recovery [50] do not support it as a tool to accelerate fatigue recovery [7,51]. Because wearing the BFR device increased perceived exhaustion in the players, other applications of BFR would need to be explored to adapt pressure and to diminish pain perception during exercise. As Kilgas et al. [37] observed, a LOP under 60% could avoid causing severe pain perception while still promoting similar results for muscle function. However, a methodology to ensure this LOP during the entire training session might not be accessible for all practitioners.

This study had some limitations. First, establishing 50 min as the minimum time to be included as a participant in this study was a great limitation because these players would start their recovery earlier than the first substitution after 5 min during the second half of the game, thus affecting the results presented here. Second, despite blinding the evaluators, the players could not be blinded to the recovery process due to the nature of the study design. Third, because we needed to incorporate a washout period between the intervention with BFR and without BFR, the players carried out the intervention at different timing in the season. To account for this factor, we randomly assigned the intervention order for all of the players. Fourth, we could not evaluate the CMJ height and RPE 48 h after the competitive soccer match because the players had a rest day. Hence, we do not know the fluctuations in these variables at this time. We only evaluated the WQ on the rest day. Fifth, because the participants were young male national-level soccer players, our results could be extrapolated to other sports population. Finally, although the players were instructed to abstain from strenuous physical exercise and to maintain their dietary habits, it was difficult to control all of the potential factors influencing perceived wellness 72 h after the soccer match. We compensated for this issue by using a crossover study design, but it should be considered in the interpretation of the results

Future studies should clarify whether active recovery methods with less pressure, load and wearing a more comfortable BFR device could help to recover neuromuscular fatigue or those parameters related to psychological effects. In addition, these recovery methodologies need to be assessed in female soccer players.

As practical applications, coaches were advised to not include active recovery sessions with BFR one day after a match because this approach has not been proved to exert a greater beneficial effect than active recovery on restoring performance, perceived exertion and wellness in youth professional soccer players. BFR could be useful if the perceived exertion needs to be immediately increased as part of competitive or compensatory training or immediately after the end of a match in order to generate greater metabolic stress. Finally, the effectiveness of this recovery protocol may be affected by the pressure, the timing in the season and the fitness level of the players.

## 5. Conclusions

The use of BFR during active recovery sessions, starting 24 h after competition, produced comparable effects to the same active recovery without BFR regarding jump ability, perceived exertion and wellness in youth male national-level soccer players. We emphasise that the use of BFR during active recovery did not show additional beneficial effects on the recovery status. In addition, BFR could induce greater immediate perceived exertion, although it would be recovered for the following training session carried out at 72 h post-match.

## Figures and Tables

**Figure 1 sports-11-00099-f001:**
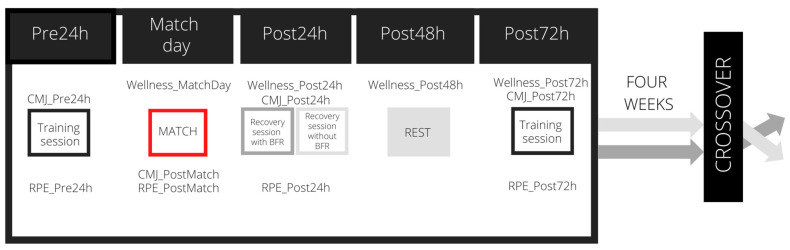
Study design. CMJ = Countermovement jump; RPE = rated of perceived exertion.

**Figure 2 sports-11-00099-f002:**
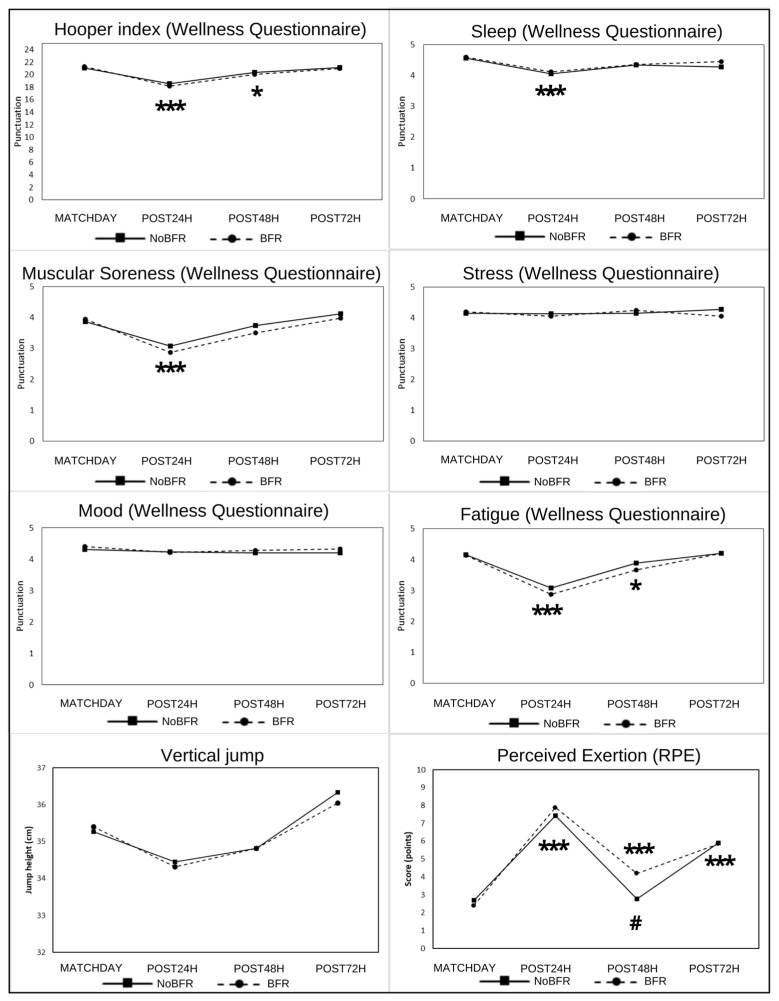
The Wellness Questionnaire, jumping ability and rating of perceived exertion measurements in the groups with blood flow restriction (BFR) and without blood flow restriction (NoBFR). The symbols indicate significant within-group differences compared with baseline (* *p* < 0.05 and *** *p* < 0.001) and between-group differences (# *p* < 0.05).

**Table 1 sports-11-00099-t001:** Countermovement jump (CMJ), rate of perceived exertion (RPE) and the Wellness Questionnaire for both conditions at baseline and all follow-up time points.

	BFR	NoBFR	Between-Group Differences	
	Mean (SD)	Mean (SD)	Mean (95% CI)	ES (d)
**CMJ (cm)**				
Pre24h	35.39 (3.62)	35.27 (3.06)	0.13 (−1.21–1.62)	NS
PostMatch	34.31 (4.01)	34.45 (4.10)	0.14 (−1.66–1.80)	NS
Post24h	34.81 (4.15)	34.81 (3.40)	0.01 (−1.70–1.71)	NS
Post72h	36.03 (4.29)	36.33 (3.30)	0.30 (−1.38–1.96)	NS
**RPE (0–10 scale)**				
Pre24h	2.40 (1.00)	2.69 (1.28)	0.29 (−0.20–0.80)	NS
PostMatch	7.87 (2.04)	7.41 (2.11)	0.46 (−0.40–1.36)	NS
Post24h	4.20 (2.11)	2.76 (1.73)	1.43 (0.56–2.30) **	0.8
Post72h	5.90 (2.1)	5.90 (1.8)	0.05 (−0.85–0.94)	NS
**Hooper index (5–25 points)**				
MatchDay	21.30 (3.08)	21.05 (2.87)	0.25 (−1.12–1.56)	NS
Post24h	18.15 (2.76)	18.56 (2.88)	0.41 (−0.85–1.60)	NS
Post48h	20.05 (2.65)	20.33 (2.65)	0.28 (−0.91–1.49)	NS
Post72h	21.00 (2.77)	21.10 (2.86)	0.10 (−1.11–1.23)	NS

BFR, blood flow restriction; CI, confidence interval; CMJ, countermovement jump; ES, effect size; NoBFR, control group; NS, not significant; SD, standard deviation. ** *p* < 0.01.

## Data Availability

The data presented in this study are available upon request from the corresponding author. The data are not publicly available due to privacy.

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
