# Peer review of "Blood Flow Restriction Is Not Useful as Soccer Competition Recovery in Youth Male National-Level Soccer Players: A Crossover Randomised Controlled Trial"

_sports, 2023, doi:10.3390/sports11050099_

Round 1

Reviewer 1 Report

Overall comment:

The Authors investigated the effects of recovery training sessions with blood flow restriction (BFR) or without it. The application of BFR during on-field sessions is novel and worth investigating.

Major revision is advised.

Specific comments:

Line 16:               Please specify which metrics were evaluated for CMJ

Line 19:               Please specify what was impaired in CMJ

Line 21:               “just the moment after 21 the recovery session” unclear, please rephrase

Line 22-24:         Please rephrase, pointing out to the act that BFR does not provide any additional benefits compared to traditional exercise modalities

Line 30:               please modify “to optimize the recovery is a crucial issue to maintain the optimal performance [1,2] and prevent more several injuries.” to “optimizing recovery is crucial to maintain optimal performance [1,2] and prevent additional injuries.”

Line 39:               please modify “train” to “trained”

Line 41-42:         “it appears…” unclear, please rephrase

Line 46:               please modify “It seems to be that” to “It seems that”

Line 49:               please modify “restoring’ to “restoration”

Line 53:               please modify “simulate” to “simulated”

Line 58:               please briefly explain why jumping ability has been chosen to assess neuromuscular function/recovery of neuromuscular function

Line 69:               why has 50 minutes been chosen as cut-off time?

Line 89:               please modify “starting the study” to “ the start of the study”

Line 91:               please modify “weight” to “body mass”

Line 92:               please check the height sd, 0.64 m seems excessive

Line 93:               please modify “During the period of the study” to “During the study period”

Line 94:               not clear, please modify “being similar for all participants”. Furthermore, can the authors provide further data – gps monitoring, sessionRPE - confirming that the two groups were exposed to comparable training loads?

Line 101:             please modify “to inform” to “to gather information”

Line 103:             please modify “ weight” to “body mass” please modify “collected in all players” to “collected for all players”

Line 107:             please modify “carried out under supervision of fitness coach, who blinded to groups of participants.” To “carried out under the supervision of the fitness coach, who was blinded the participants’ group assignment.”

Line 110:             what does “on standing position” refers to? Please explain or modify.
please include more detail on the verbal cues and indications provided to the athletes, please refer to https://doi.org/10.3390/sports8090127

Line 111:             please modify “Three jumps separately were performed” to “Three non-consecutive jumps were performed”

Line 112:             is the force plate used the same as line 123? Please include the model of the force plate used.

Line 113:             why jump height was the only investigated variable? Please refer to https://doi.org/10.1519/SSC.0000000000000677

Line 117:             please modify “competitive” to “competition”

Line 121:             please modify “the match day” to “at match day”

Line 122:             please remove “of the platform”

Line 124-125:     unclear, please rephrase

Line 128:             please modify “the training” to “the end of the training”

Line 134:             please remove “as”

Line 136;             please modify “during” to “in”

Line 140:             how long passed between the familiarization session and the first intervention?

Line 146:             what does “manually” refer to? “The standing position were used” is unclear, please modify

Line 148:             please modify “was determinate” to “was determinated”

                             Please modify “next” to “following”

Line 152:             60-70% refers to a self-selected pace or was it assessed in respect to the athletes maximal sprint speed?

Line 154:             Please provide additional details on the rounds: number of sets, durations of each set, number of players involved.

Line 155:             please modify “A supplementary Figure S1 shows an example of these exercises.” To “In the supplementary Figure S1 is represented an example of these exercises.”

Line 169:             please modify “Table 1 shows results of RPE, CMJ and WQ” to “RPE, CMJ, and WQ are shown in Table 1”

Line 174:             please modify “ in RPE” to “Concerning RPE”

Line 196:             please remove “similarly”

Line 197:             please remove “being”

Line 205:             please modify “Data from sub-items are presented” to “ Data from sub-items is presented”

Line 212:             please modify “showed” to “is”

Line 214:             please modify “wellness perception” to “perceived wellness”

Line 219-220:    wordy, please rephrase

Line 221:             please provide context for the use of the term “cyclically”

 Line 224:            please modify “evaluated effects” to ” evaluated the effects”

Line 225:             Please modify “of” to “in”

Line 228:             please expand on the key factors that could condition the beneficial effects.

Line 230:             please modify “for motivation of players” to “to motivate players”

Line 232:             please remove “, whose”

Line 233:             please modify “of” to “in”

Line 235-237:    “Although we observed an immediate increase of 235 RPE when players wore BFR, in our study, this negative affection was not observed for 236 the following measurements or the rest of variables.” Unclear, please rephrase

Line 238:             please modify “cannot observed” to “could not observe”

Line 240:             please modify “refers to” to “is”

Line 243:             please modify “expectable” to “expected”

                             Please modify “to the fact that” to “because”

Line 244:             what do the Authors mean by saying that RPE was used to carry out a hard session? Please clarify

Line 248:             please modify “high volume” to “congestion”

Line 249:             please modify “ability of jumping” to “jumping ability”

Line 250:             please remove ” of our study”
please add “movement onwards”

Line 256:             please modify “regardless the condition” to “regardless of the condition”

Line 258:             please modify “vertical jump recovery” to “recovery in vertical jump ability”

Line 260:             is unclear what the authors refer to with ”punctuations”, please clarify

Line 265:             please modify “did” to “was”

Line 266:             please modify “on individual characteristics of players, by suggesting” to “on the individual characteristics of the players, suggesting”

Line 269-270:     please avoid writing in first person

Line 275:             please modify ‘academic” to “academy-level”

                             Please modify “in line” to “Likewise”

Line 275:             please avoid writing in first person

Line 276:             please modify “that most lasted to be completely recovered” to “that took the longest to be completely recovered”

Line 277:             “As difference to note” is unclear, please rephrase

                             Who “authors” refers to? Please provide reference

Line 281:             please modify “fatigue perception” to “perception of fatigue”

please avoid writing in first person

Line 282:             please remove “to be”

Line 285:             please modify “it would be needed to explore other BFR applications” to “other applications of BFR would need to be explored”

Line 287:             please modify “to cause” to “causing”

Line 291:             please avoid writing in first person

Line 293:             please modify “order of the intervention were” to “ intervention order was”

Line 294:             please avoid writing in first person

Line 304:             the design of the study does not support the use of recovery sessions regardless of the intervention, in fact no control group not carrying out any recovery activity 24 hours after the game was present. Therefore authors should avoid claiming the efficacy of recovery sessions.

Line 306:             please clarify why perceived exertion should be increased

Line 311:             please modify “  is similar” to “carries comparable effects”

Line 312:             please avoid writing in first person

Line 313-314:     please refrain from claiming that the exhaustion would be recovered for the following session, as the study design did not assess it 48h post match.

The manuscript would benefit from extensive language review as part of it are unclear.

Author Response

Dear Reviewer,

Thank you for your helpful comments and your proposal to resubmit our paper entitled: “Effect of Blood Flow Restriction as Soccer Competition Recovery in Youth Male National Level Soccer Players: A Randomized Crossover Controlled Trial”, pending the correction of these some major issues.

Authors are very grateful for the useful comments and time spent. The modifications have been marked up using the “Track Changes” function such that any changes can be easily located. Also, the replies to reviewer' comments are included, point by point, by explaining the details of the revisions, in bold and red format, in the attached document.

We thank you for your consideration and hope that our responses will come up to your expectations.

Yours sincerely,

The authors

The Authors investigated the effects of recovery training sessions with blood flow restriction (BFR) or without it. The application of BFR during on-field sessions is novel and worth investigating.

Major revision is advised.

Specific comments:

Line 16:               Please specify which metrics were evaluated for CMJ

We considered height of the jump. Now, metrics for CMJ have been added.

Line 19:               Please specify what was impaired in CMJ

We have added “jump height” for “CMJ” abbreviation in previous lines to clear this aspect.

Line 21:               “just the moment after the recovery session” unclear, please rephrase

Since the recovery session was completed by players the following day after the soccer match, “24 hours later” is also referring to the time immediately after the recovery session. Due to the complex study protocol, Figure 1 may help understand it. It has been modified in abstract.

Line 22-24:         Please rephrase, pointing out to the act that BFR does not provide any additional benefits compared to traditional exercise modalities

It has been modified and this aspect has been highlighted.

Line 30:               please modify “to optimize the recovery is a crucial issue to maintain the optimal performance [1,2] and prevent more several injuries.” to “optimizing recovery is crucial to maintain optimal performance [1,2] and prevent additional injuries.”

Modified.

Line 39:               please modify “train” to “trained”

Modified.

Line 41-42:         “it appears…” unclear, please rephrase

Rephrased.

Line 46:               please modify “It seems to be that” to “It seems that”

Modified.

Line 49:               please modify “restoring’ to “restoration”

Modified.

Line 53:               please modify “simulate” to “simulated”

Modified.

Line 58:               please briefly explain why jumping ability has been chosen to assess neuromuscular function/recovery of neuromuscular function

Jumping ability (CMJ) has been chosen due to the sensitivity, reliability and validity of this task to detect neuromuscular fatigue in team sports, according to previous studies (21). It has been added to the text: (Line 305-310)

Line 69:               why has 50 minutes been chosen as cut-off time?

It was considered because normally the first substitutions usually occur between the 60 and 70 minutes of a soccer match. Although it is true that after COVID19, five substitutions have been authorized in three interruptions in professional soccer players. Consequently, these data have been varying with respect to when the first substitutions occur. On the other hand, in the Youth Male National Level Soccer Players (Youth Honor Spanish division league), up to a maximum of seven substitution can be made in four interruptions. In addition, all the players analyzed in this study were in the starting line-up, and played (79.85 ± 14.06 min). The minutes played has been added in participants section (Line 111-112).

Line 89:               please modify “starting the study” to “ the start of the study”

Modified.

Line 91:               please modify “weight” to “body mass”

Modified.

Line 92:               please check the height sd, 0.64 m seems excessive

It was a mistake. It has been checked.

Line 93:               please modify “During the period of the study” to “During the study period”

Modified.

Line 94:               not clear, please modify “being similar for all participants”.

Modified.

Furthermore, can the authors provide further data – gps monitoring, session RPE - confirming that the two groups were exposed to comparable training loads?

In order to control the comparable training load (TL) between the two groups was evaluated the RPE and multiplied by the duration of the training session according to other studies. At 24h (BFR condition: 187.03 ± 92.53 a.u; NonBFR condition: 140.51 ± 64.23 a.u.) and at 72h ( BFR condition: 505.73 ± 191.32 a.u ;NonBFR condition: 514.51 ± 208.39 a.u). Additionally, the external load was measure by GPs Data wearing during the training session and matches. It has been added to the text. (Line 164-168)

Line 101:             please modify “to inform” to “to gather information”

Modified.

Line 103:             please modify “ weight” to “body mass” please modify “collected in all players” to “collected for all players”

Modified.

Line 107:             please modify “carried out under supervision of fitness coach, who blinded to groups of participants.” To “carried out under the supervision of the fitness coach, who was blinded the participants’ group assignment.”

Modified.

Line 110:             what does “on standing position” refers to? Please explain or modify.
please include more detail on the verbal cues and indications provided to the athletes, please refer to https://doi.org/10.3390/sports8090127

Detailed information has been added: line 134-136

Line 111:             please modify “Three jumps separately were performed” to “Three non-consecutive jumps were performed”

Modified.

Line 112:             is the force plate used the same as line 123? Please include the model of the force plate used.

Detailed information has been added: line 139

Line 113:             why jump height was the only investigated variable? Please refer to https://doi.org/10.1519/SSC.0000000000000677

Due to the complex protocol of study and the close measurement times, only one performance variable was added to avoid extending the time to evaluate. Additionally, the jump heigh is one of the most evaluated metrics nowadays. It has been added to the text and reference has been added. Line 139-143

Line 117:             please modify “competitive” to “competition”

Modified.

Line 121:             please modify “the match day” to “at match day”

Modified.

Line 122:             please remove “of the platform”

Modified.

Line 124-125:     unclear, please rephrase

Rephrased.

Line 128:             please modify “the training” to “the end of the training”

Modified.

Line 134:             please remove “as”

Modified.

Line 136;             please modify “during” to “in”

Modified.

Line 140:             how long passed between the familiarization session and the first intervention?

Two weeks for all players. It has been added to the text.

Line 146:             what does “manually” refer to? “The standing position were used” is unclear, please modify

This aspect has been clarified and modified in the text.

Line 148:             please modify “was determinate” to “was determinated”

Modified.

                             Please modify “next” to “following”

Modified.

Line 152:             60-70% refers to a self-selected pace or was it assessed in respect to the athletes maximal sprint speed?

It was measured in respect to the maximum sprint speed in each soccer players. The 30m sprint test was executed during preseason for all the players. This information has been added to the text. (Interventions section) Line 195-200.

Line 154:             Please provide additional details on the rounds: number of sets, durations of each set, number of players involved.

Detailed information has been added for the exercises: line 195-202

Line 155:             please modify “A supplementary Figure S1 shows an example of these exercises.” To “In the supplementary Figure S1 is represented an example of these exercises.”

Modified.

Line 169:             please modify “Table 1 shows results of RPE, CMJ and WQ” to “RPE, CMJ, and WQ are shown in Table 1”

Modified.

Line 174:             please modify “ in RPE” to “Concerning RPE”

Modified.

Line 196:             please remove “similarly”

Modified.

Line 197:             please remove “being”

Modified.

Line 205:             please modify “Data from sub-items are presented” to “ Data from sub-items is presented”

Modified.

Line 212:             please modify “showed” to “is”

Modified.

Line 214:             please modify “wellness perception” to “perceived wellness”

Modified.

Line 219-220:    wordy, please rephrase

This sentence has been modified.

Line 221:             please provide context for the use of the term “cyclically”

It has been detailed in the text: Line 268-272.

Line 224:            please modify “evaluated effects” to ” evaluated the effects”

Modified.

Line 225:             Please modify “of” to “in”

Modified.

Line 228:             please expand on the key factors that could condition the beneficial effects.

This aspect has been extended: lines 275-282

Line 230:             please modify “for motivation of players” to “to motivate players”

Modified.

Line 232:             please remove “, whose”

Modified.

Line 233:             please modify “of” to “in”

Modified.

Line 235-237:    “Although we observed an immediate increase of 235 RPE when players wore BFR, in our study, this negative affection was not observed for 236 the following measurements or the rest of variables.” Unclear, please rephrase

Rephrased.

Line 238:             please modify “cannot observed” to “could not observe”

Modified.

Line 240:             please modify “refers to” to “is”

Modified.

Line 243:             please modify “expectable” to “expected”

Modified.

                             Please modify “to the fact that” to “because”

Modified.

Line 244:             what do the Authors mean by saying that RPE was used to carry out a hard session? Please clarify

We meant that the high RPE values is according to the training load distribution during the week. It has been clarified in the text: lines 298-301

Line 248:             please modify “high volume” to “congestion”

Modified.

Line 249:             please modify “ability of jumping” to “jumping ability”

Modified.

Line 250:             please remove ” of our study”

Removed.

Line 250: please add “movement onwards”

We do not understand to which part of 250 line reviewer proposes to add “movement onwards”. Could you specify it?

Line 256:             please modify “regardless the condition” to “regardless of the condition”

Modified.

Line 258:             please modify “vertical jump recovery” to “recovery in vertical jump ability”

Modified.

Line 260:             is unclear what the authors refer to with ”punctuations”, please clarify

It has been clarified.

Line 265:             please modify “did” to “was”

Modified.

Line 266:             please modify “on individual characteristics of players, by suggesting” to “on the individual characteristics of the players, suggesting”

Modified.

Line 269-270:     please avoid writing in first person

We have modified first person in the entire manuscript, as suggested.

Line 275:             please modify ‘academic” to “academy-level”

Modified.

                             Please modify “in line” to “Likewise”

Modified.

Line 275:             please avoid writing in first person

We have modified first person in the entire manuscript, as suggested.

Line 276:             please modify “that most lasted to be completely recovered” to “that took the longest to be completely recovered”

Modified.

Line 277:             “As difference to note” is unclear, please rephrase

It has been rephrased.

                             Who “authors” refers to? Please provide reference

A reference has been added.

Line 281:             please modify “fatigue perception” to “perception of fatigue”

Modified.

please avoid writing in first person

We have modified first person in the entire manuscript, as suggested.

Line 282:             please remove “to be”

Removed.

Line 285:             please modify “it would be needed to explore other BFR applications” to “other applications of BFR would need to be explored”

Modified.

Line 287:             please modify “to cause” to “causing”

Modified.

Line 291:             please avoid writing in first person

We have modified first person in the entire manuscript, as suggested.

Line 293:             please modify “order of the intervention were” to “ intervention order was”

Modified.

Line 294:             please avoid writing in first person

We have modified first person in the entire manuscript, as suggested.

Line 304:             the design of the study does not support the use of recovery sessions regardless of the intervention, in fact no control group not carrying out any recovery activity 24 hours after the game was present. Therefore authors should avoid claiming the efficacy of recovery sessions.

The reviewer is right. We have adapted this information in the text. Lines 387-390

Line 306:             please clarify why perceived exertion should be increased

It has been clarified.

Line 311:             please modify “  is similar” to “carries comparable effects”

Modified.

Line 312:             please avoid writing in first person

We have modified first person in the entire manuscript, as suggested.

Line 313-314:     please refrain from claiming that the exhaustion would be recovered for the following session, as the study design did not assess it 48h post match.

We referred to the following training session that was scheduled in our study, which was established at Post72h (because the resting session was established at Post48h). It has been clarified in the text. Line 398-402

Reviewer 2 Report

Dear Authors,

You have written an interesting paper focusing on the effects of BFR as a recovery strategy after a competition in jump, perceived exertion and wellness of soccer players.

However, there are several parts that need to be addressed for greater clarity.

In the introduction, I don't see how previous research on IPC affected recovery in other sports and soccer-team sports, which could be the foundation for your research and soccer. Additionally, it would add an in-depth explanation to your discussion.

Here are some papers that could be useful for your paper:

-https://doi.org/10.1080/02640414.2014.988741

-10.1016/j.jsams.2021.02.012

- https://doi.org/10.1123/ijspp.2022-0280

Methods:

Figure 1 is of poor visibility. Try to use just the first part, as the second one is just the repeat.

Well done on using G*Power.

Line 69 - why just 50min? So technically someone could be substituted after 5 minutes of the second half and be included in your study. His recovery starts much sooner than players finishing the match. Please elaborate as this is a great limitation of your study.

Inclusion criteria should be better defined - training experience, matches played, etc

What were their playing positions? Did you include goalkeepers in the protocol? report

Line 105 - and what was that similar daytime? report

Line 111 - back up your break of 45s with a reference.

Line 119 - report the speed and model of the treadmill.

What CMJ variables were measured and what result from 3 trials was taken into further analysis? report

A reference of validity for the WQ questionnaire needs to be reported.

The exact place where the cuffs were placed is not reported. Add

Line 152-173 - How did you monitor the intensity of 60-70% of Max.

What activation part? If this is the S1 picture then this is a very poor activation exercise!

Line 155 - there is just 1 exercise on the S1 picture that is not presenting the exercise really well. Amend

The BFR recovery session is very unclear- So were the cuffs applied on both legs at the same time; for how long were they inflated before having a 90s deflation; etc. From this description, you can't replicate your study.

The discussion is solid, however, the proposed recovery studies from team and other sports should be included.

Limitations of the study should be expanded as the inclusion criteria of min 50 min of playtime is an essential factor that could affect your results and your recovery protocol.

Overall an interesting study, but lacks methodological precision. However, I would still like to give authors a chance to improve their paper. Therefore, I recommend a major revision.

Best regards

Minor editing of English language required

Author Response

Dear Reviewer,

Thank you for your helpful comments and your proposal to resubmit our paper entitled: “Effect of Blood Flow Restriction as Soccer Competition Recovery in Youth Male National Level Soccer Players: A Randomized Crossover Controlled Trial”, pending the correction of these some major issues.

Authors are very grateful for the useful comments and time spent. The modifications have been marked up using the “Track Changes” function such that any changes can be easily located. Also, the replies to reviewer' comments are included, point by point, by explaining the details of the revisions, in bold and red format, in the attached document.

We thank you for your consideration and hope that our responses will come up to your expectations.

Yours sincerely,

The authors

Dear Authors,

 You have written an interesting paper focusing on the effects of BFR as a recovery strategy after a competition in jump, perceived exertion and wellness of soccer players. However, there are several parts that need to be addressed for greater clarity.

We thank reviewer the supportive comment. I have tried to address all parts to improve clarity, according to each comment.

In the introduction, I don't see how previous research on IPC affected recovery in other sports and soccer-team sports, which could be the foundation for your research and soccer.

According to the recommendation, the foundation of our research and how IPC may affect recovery in other soccer-team sports have been extended: lines 50-66

Additionally, it would add an in-depth explanation to your discussion.

This aspect has been also extended in the Discussion section.

Here are some papers that could be useful for your paper:

-https://doi.org/10.1080/02640414.2014.988741

-10.1016/j.jsams.2021.02.012

- https://doi.org/10.1123/ijspp.2022-0280

Methods:

Figure 1 is of poor visibility. Try to use just the first part, as the second one is just the repeat.

Figure 1 has been modified and only the first part has been considered to improve the visibility.

Well done on using G*Power.

Line 69 - why just 50min? So technically someone could be substituted after 5 minutes of the second half and be included in your study. His recovery starts much sooner than players finishing the match. Please elaborate as this is a great limitation of your study.

It was considered because normally the first substitutions usually occur between the 60 and 70 minutes of a soccer match. Although it is true that after COVID19, five substitutions have been authorized in three interruptions in professional soccer players. Consequently, these data have been varying with respect to when the first substitutions occur. On the other hand, in the Youth Male National Level Soccer Players (Youth Honor Spanish division league), up to a maximum of seven substitution can be made in four interruptions. In addition, all the players analyzed in this study were in the starting line-up, and played (79.85 ± 14.06 min). The minutes played has been added in participants section (Line 111-112).

The limitation factor has been added in the text: lines 366-369

Inclusion criteria should be better defined - training experience, matches played, etc

Inclusion criteria has been detailed: lines 101-105

What were their playing positions? Did you include goalkeepers in the protocol? report

We registered their playing positions and this aspect has been included in the Method section. Goalkeepers were excluded because the fatigue is not comparable fatigue with the rest of the players positions (defences, midfielders and forwards) It has been explained in the text: lines 105-107

Line 105 - and what was that similar daytime? report

The daytime has been added: line 125

Line 111 - back up your break of 45s with a reference.

A reference has been included.

Line 119 - report the speed and model of the treadmill.

The speed and model of the treadmill have been included: lines 150

What CMJ variables were measured and what result from 3 trials was taken into further analysis? report a reference of validity for the WQ questionnaire needs to be reported.

We considered the jump height of the CMJ as variable after taking into consideration the average of these three trials. It has been included in the text: lines 139-143

A reference of validity of WQ has been added. (McLean et al., 2010) and (Hooper et al., 1995)

The exact place where the cuffs were placed is not reported. Add

The reviewer is right, this information has been added now: lines 189-190

Line 152-173 - How did you monitor the intensity of 60-70% of Max.

We monitored the intensity with the time derived and calculated from the 30 m sprint test evaluated during preseason. This information has been added: lines 195-200

What activation part? If this is the S1 picture then this is a very poor activation exercise!

Figure S1 is only an example, but players performed several exercises as activation part. Other exercises were executed prior to beginning of the recovery session. This information has been added: lines 200-202

Line 155 - there is just 1 exercise on the S1 picture that is not presenting the exercise really well. Amend

The information has been extended in the Methods section.

The BFR recovery session is very unclear- So were the cuffs applied on both legs at the same time; for how long were they inflated before having a 90s deflation; etc. From this description, you can't replicate your study.

The BFR recovery session has been detailed to make our study replicable: lines 185-202

The discussion is solid, however, the proposed recovery studies from team and other sports should be included.

According the recommendation, this information has been included in discussion section.

Limitations of the study should be expanded as the inclusion criteria of min 50 min of playtime is an essential factor that could affect your results and your recovery protocol.

 We agree with the reviewer and limitations have been expanded to acknowledge this limitation of the study: lines 366-369

Overall an interesting study, but lacks methodological precision. However, I would still like to give authors a chance to improve their paper. Therefore, I recommend a major revision.

We thank reviewer for the opportunity to improve our paper.

Reviewer 3 Report

- Line 29: 3 times? please indicate where? Is it worldwide? I believe not.

- Line 37: needs citation to support your claim.

- Line 43: again, according to who? the authors describe several bold statements without referencing them.

- Dabb et al. need citation.

- Overall, the introduction section failed to convince me that this study is needed. There are no hypotheses, no discussion of previous studies using BFR comparing it with other recovery methods (e.g., ice baths, passive stretching, placebo effects), and what could be the practical applications of the present study. Without this, this study feels like a research note.

- Line 74: How were participants randomly assigned? which method did the authors use?

- Line 84: Report all parameters for sample size calculations.

- report detailed data collection procedures and disclose how many teams were contacted, how the authors contacted clubs, etc. This needs to be disclosed for replication purposes.

- Lines 126-130: One of the biggest issues in this manuscript is the measurement of perceived effort. Why measure PE 30 minutes after the stimulus and not immediately after? In addition, measure Pre24h, Post-Match, Post24h, and Post72h are not clear since there are so many confounding variables that could influence PE post72h? Thus, all measurements seem to be related to covariates (the authors report 3 games per week) and not the task itself.

- The Hooper index does not measure well-being. It measures stimulus-induced fatigue. Measuring it Post72h has no fundamental need since the fatigue after Post72h is difficult to assume as match-induced fatigue.

- Line 143: which limbs? clarify.

- Line 159: K-S test should be applied in samples n>50.

- Lines 161: how did the authors control for covariate variables (experience, effort during the match) in ANOVA analyses?

- In addition, repeated measures ANOVA should be accompanied by the Greenhaus-Geisser adjusted values and degrees of freedom test, and assume the assumption of sphericity using Mauchly's test should be reported.

Grammar errors and syntax writing needs revisions.

Author Response

Dear Reviewer,

Thank you for your helpful comments and your proposal to resubmit our paper entitled: “Effect of Blood Flow Restriction as Soccer Competition Recovery in Youth Male National Level Soccer Players: A Randomized Crossover Controlled Trial”, pending the correction of these some major issues.

Authors are very grateful for the useful comments and time spent. The modifications have been marked up using the “Track Changes” function such that any changes can be easily located. Also, the replies to reviewer' comments are included, point by point, by explaining the details of the revisions, in bold and red format, in the attached document.

We thank you for your consideration and hope that our responses will come up to your expectations.

Yours sincerely,

The authors

- Line 29: 3 times? please indicate where? Is it worldwide? I believe not.

Up to three matches per week is the maximal matches that players reached. This occurs in Spain and the rest of European countries and only in certain periods of the season, but is more and more frequent each season due to the social demands and soccer monetization. This information has been extended to improve clarity.

- Line 37: needs citation to support your claim.

A reference has been added.

- Line 43: again, according to who? the authors describe several bold statements without referencing them.

References have been added.

- Dabb et al. need citation.

The reference has been included.

- Overall, the introduction section failed to convince me that this study is needed. There are no hypotheses, no discussion of previous studies using BFR comparing it with other recovery methods (e.g., ice baths, passive stretching, placebo effects), and what could be the practical applications of the present study. Without this, this study feels like a research note.

According to the recommendation, the foundation of our research and how IPC may affect recovery in other soccer-team sports have been extended: lines 50-66

 Also, potential practical applications of the study have been added: lines 67-70

- Line 74: How were participants randomly assigned? which method did the authors use?

We used a web-based software (randomize.org). It has been added to the text: line 89

- Line 84: Report all parameters for sample size calculations.

It has been detailed: lines 97-100

- report detailed data collection procedures and disclose how many teams were contacted, how the authors contacted clubs, etc. This needs to be disclosed for replication purposes.

All regional soccer teams that participated in the Spanish Youth Soccer League (n = 6) were contacted by mailing to the team staff. Only three soccer team accepted to participate in the study. This information has been extended: lines 107-109

- Lines 126-130: One of the biggest issues in this manuscript is the measurement of perceived effort. Why measure PE 30 minutes after the stimulus and not immediately after? In addition, measure Pre24h, Post-Match, Post24h, and Post72h are not clear since there are so many confounding variables that could influence PE post72h? Thus, all measurements seem to be related to covariates (the authors report 3 games per week) and not the task itself.

We understand the comment, however, PE is frequently assessed 30 minutes after soccer training and competitive soccer matches by most of the sports teams in order to ensure that the perceived effort was referred to the whole season rather than the most recent intensity. The reason why our study also assessed PE 30 minutes after the stimulus is to maintain the coherence with the rest of evaluations in the team and also in agreement with similar previous studies (27).

- The Hooper index does not measure well-being. It measures stimulus-induced fatigue. Measuring it Post72h has no fundamental need since the fatigue after Post72h is difficult to assume as match-induced fatigue.

According to the reviewer suggestion, we have modified the interpretation of Hooper index according to the aspect that this variable is referring (lines 167-169).

Also, we agree with the reviewer that the reason of the fatigue at Post72h may be affected by other aspects. To try to compensate this aspect, we gave players instructions to abstained from strenuous activity outside of the proposed training or competition during the period of intervention (from 24h before testing to Post72h measurement) and were encouraged to maintain their normal dietary and fluid intake habits during the research. This information has been extended to Method section (lines 126-129). It has been also included as an aspect to consider in the limitations of the study: lines 366-369

- Line 143: which limbs? clarify.

This information and details regarding BFR protocol have been added now: lines 185-202

- Line 159: K-S test should be applied in samples n>50.

Reviewer is right. Shapiro-Wilk test has been applied now. It has been modified in the text and results.

- Lines 161: how did the authors control for covariate variables (experience, effort during the match) in ANOVA analyses?

We did not add covariate variables because there were no significant differences at baseline among players. To monitor effort during the match and training session, training load of players was monitored by GPS technology (distance covered during the match and training sessions, in meters) and RPE x duration of the session training expressed in A.U. Values were similar for all the players included in the study. This information has been added to the Method section.

Since it was difficult to control all aspects related to subjective variables, procedures were completed by following similar methodologies than previous studies (Daab et al., 2021; Deely et al., 2022; Page et al., 2017).

Daab, W., Bouzid, M. A., Lajri, M., Bouchiba, M., & Rebai, H. (2021). Brief cycles of lower-limb occlusion accelerate recovery kinetics in soccer players. The Physician and Sportsmedicine, 49(2), 143-150.

Deely, C., Tallent, J., Bennett, R., Woodhead, A., Goodall, S., Thomas, K., & Howatson, G. (2022). Etiology and recovery of neuromuscular function following academy soccer training. Frontiers in Physiology, 1170.

Page, W., Swan, R., & Patterson, S. D. (2017). The effect of intermittent lower limb occlusion on recovery following exercise-induced muscle damage: A randomized controlled trial. Journal of science and medicine in sport, 20(8), 729-733.

- In addition, repeated measures ANOVA should be accompanied by the Greenhaus-Geisser adjusted values and degrees of freedom test, and assume the assumption of sphericity using Mauchly's test should be reported.

According to the recommendation, statistical analysis have been modified after using the proposed procedures. Results were not different after applying these statistical analysis for any variable.

Round 2

Reviewer 1 Report

Overall comment:

The Authors apported considerable changes that elevated the overall quality of the manuscript.

The readability of the manuscript has been improved by the Authors.

Only minor revisions are needed, please see specific comments.

Specific comments:

Line 94:               please modify “placed” to “place”

Line 221:             please refrain from writing in first person

Line 287-288:     please refrain from writing in first person

Line 299-300:     “increasing nociceptive group IV muscle afferents nociceptive” unclear, please rephrase

Line 303:             please modify “suggest” to “have suggested”

Line 309:             please modify “academic” to “academy”

Line 323:             please refrain from writing in first person

Line 325:             please refrain from writing in first person

Line 327:             please refrain from writing in first person

Line 381:             please refrain from writing in first person

Line 419:             “including drills” not clear, please rephrase

Author Response

Dear Reviewer,

Once again, thank you for your new comments and your proposal to resubmit our paper “Effect of Blood Flow Restriction as Soccer Competition Recovery in Youth Male National Level Soccer Players: A Crossover Randomised Controlled Trial”, pending the correction of these some minor issues.

Authors are very grateful for the useful comments and time spent. The modifications have been marked up using the “Track Changes” function such that any changes can be easily located. Also, the replies to reviewer' comments are included, point by point, by explaining the details of the revisions, in bold format, in the attached document.

We thank you for your consideration and hope that our responses will come up to your expectations.

Yours sincerely,

The authors

The Authors apported considerable changes that elevated the overall quality of the manuscript.

The readability of the manuscript has been improved by the Authors.

Only minor revisions are needed, please see specific comments.

 Specific comments:

Line 94:               please modify “placed” to “place”

Modified.

Line 221:             please refrain from writing in first person

Checked in the entire paragraph.

Line 287-288:     please refrain from writing in first person

Checked.

Line 299-300:     “increasing nociceptive group IV muscle afferents nociceptive” unclear, please rephrase

This sentence had a mistake. It has been corrected.

Line 303:             please modify “suggest” to “have suggested”

Modified.

Line 309:             please modify “academic” to “academy”

Modified.

Line 323:             please refrain from writing in first person

Checked in the entire paragraph.

Line 325:             please refrain from writing in first person

Checked in the entire paragraph.

Line 327:             please refrain from writing in first person

Checked in the entire paragraph.

Line 381:             please refrain from writing in first person

Checked.

Line 419:             “including drills” not clear, please rephrase

This part has been deleted.

Reviewer 2 Report

Dear Authors,

Thank you for a very good job in answering my questions and recommendations. The authors did a very good job, and the paper quality increased accordingly to the effort put into revision!

Therefore, I recommend acceptance in its current form.

Kind regards 

 Minor editing of the English language required

Author Response

Dear Reviewer,

Once again, thank you for your new comments and your proposal to resubmit our paper “Effect of Blood Flow Restriction as Soccer Competition Recovery in Youth Male National Level Soccer Players: A Crossover Randomised Controlled Trial”, pending the correction of these some minor issues.

Authors are very grateful for the useful comments and time spent. The modifications have been marked up using the “Track Changes” function such that any changes can be easily located. Also, the replies to reviewer' comments are included, point by point, by explaining the details of the revisions, in bold format, in the attached document.

Yours sincerely,

The authors

Dear Authors,

Thank you for a very good job in answering my questions and recommendations. The authors did a very good job, and the paper quality increased accordingly to the effort put into revision!

Therefore, I recommend acceptance in its current form.

Kind regards

We thank reviewer the supportive comment and the decision made on our manuscript.

Reviewer 3 Report

The authors did a good job reviewing their manuscript. I have no further comments.

None.

Author Response

Dear Reviewer,

Once again, thank you for your new comments and your proposal to resubmit our paper “Effect of Blood Flow Restriction as Soccer Competition Recovery in Youth Male National Level Soccer Players: A Crossover Randomised Controlled Trial”, pending the correction of these some minor issues.

Authors are very grateful for the useful comments and time spent. The modifications have been marked up using the “Track Changes” function such that any changes can be easily located. Also, the replies to reviewer' comments are included, point by point, by explaining the details of the revisions, in bold format, in the attached document.

Yours sincerely,

The authors

The authors did a good job reviewing their manuscript. I have no further comments.

We thank reviewer the supportive comment and the decision made on our manuscript.